# Intranasal Dexmedetomidine Compared to a Combination of Intranasal Dexmedetomidine with Ketamine for Sedation of Children Requiring Dental Treatment: A Randomized Clinical Trial

**DOI:** 10.3390/jcm10132840

**Published:** 2021-06-27

**Authors:** Joji Sado-Filho, Patrícia Corrêa-Faria, Karolline A. Viana, Fausto M. Mendes, Keira P. Mason, Luciane R. Costa, Paulo S. Costa

**Affiliations:** 1Health Sciences Graduate Program, Faculdade de Medicina, Universidade Federal de Goiás, Goiânia 74605-050, Brazil; joji.sado@gmail.com (J.S.-F.); lsucasas@ufg.br (L.R.C.); 2Dentistry Graduate Program, Faculdade de Odontologia, Universidade Federal de Goiás, Goiânia 74605-220, Brazil; patriciacorreafaria@ufg.br (P.C.-F.); karollineav@gmail.com (K.A.V.); 3Dentistry Graduate Program, Faculdade de Odontologia, Universidade de São Paulo, São Paulo 05508-000, Brazil; fmmendes@usp.br; 4Department of Anesthesiology, Critical Care and Pain Medicine, Harvard Medical School, Boston Children’s Hospital, Boston, MA 02115, USA; Keira.Mason@childrens.harvard.edu

**Keywords:** moderate sedation, dexmedetomidine, ketamine, intranasal administration, dental care for children, dental anxiety

## Abstract

Outpatient pediatric sedation is challenging. This study aimed to test intranasal dexmedetomidine efficacy as a single drug or combined with ketamine (DK) to sedate children undergoing dental treatment. Children < 7 years were randomized into dexmedetomidine 2 mcg/kg and ketamine 1 mg/kg (DK) or dexmedetomidine 2.5 mcg/kg (D) groups. Videos from the dental sedation allowed the systematic assessment of children’s behavior (primary outcome) according to the Ohio State University Behavioral Rating Scale (OSUBRS). Secondary outcomes were parental and dentist satisfaction, adverse events, and recovery time. The data were analyzed descriptively and through regression models. Participants were 88 children (44 per group; 50 boys). The duration of quiet behavior (OSUBRS) was higher than 50% (DK mean 58.4 [standard deviation 38.1]; D 55.2 [39.1]; *p* = 0.225). Parents (DK 78.0 [32.2]; D 72.7 [35.1]; *p* = 0.203) and dentists (KD 62.7 [41.0]; D 62.8 [40.1]; *p* = 0.339) were overall satisfied. Adverse events occurred in 16 cases (DK *n* = 10, 62.5%; D *n*= 6, 37.5%; *p* = 0.104) and were minor. The median recovery time in the DK group was 1.3 times greater than in group D (*p* < 0.05). Intranasal sedation with dexmedetomidine alone is equally efficacious and satisfactory for pediatric sedation with fewer adverse events and faster recovery than the DK combination.

## 1. Introduction

Approximately four in 10 children have dental fear/anxiety [1]. In some cases, the anxiety level is combined with dental behavioral management problems (DBMP) and impedes the child’s ability to tolerate routine, unsedated dental treatment [2]. Dental anxiety and DBMP lead to an increased duration of dental procedures and elevated costs to the patient and the dental provider [3]. Children with extreme dental fear/anxiety, or neuro-cognitive or behavioral disabilities, are most frequently in need of sedation [2].

There are a variety of sedatives for pediatric sedation [4]. Chloral hydrate [5] and benzodiazepines [6] have historically been the mainstay of pediatric dental sedation in combination or not with antihistaminics, meperidine, or other pharmacological agents [7,8]. These sedation regimens are limited by routes of administration, long half-lives, higher risk of respiratory depression and apnea, or discontinuation of production [9,10]. Besides, chloral hydrate lacks drug products approved by regulatory agencies such as the Food and Drug Administration [11]. Ketamine has been described for pediatric dental sedation, with administration in different routes such as oral [12,13,14], subcutaneous, intramuscular injection, intravenously [14], and intranasal [12,13]. Ketamine offers the advantage of analgesia [15], amnesia [16], and hemodynamic stability [17] with the associated risks of apnea, respiratory compromise, hallucinations, transient laryngospasm (rare), excitatory behavior, and vomiting (more common, 5 to 15%) [17,18,19]. The potential of dexmedetomidine to attenuate the sympathetic response, provide sedation, neuroprotection, and decrease emergence agitation are properties that may be favorable in its combination with ketamine [20]. Recent literature supports that dexmedetomidine provides a synergy with ketamine, which would be advantageous in enabling a decrease in dosing of both sedatives [21,22].

The child’s behavior is one of the outcomes evaluated in studies on the efficacy of sedatives in dental procedures. Previous investigations showed that the efficacy of dexmedetomidine is similar to that of midazolam and ketamine administered both intranasally [15] and orally [23]. There is little evidence about the effect of the combination of dexmedetomidine and ketamine on the behavior of children undergoing dental treatment [24].

This clinical trial aimed to evaluate the efficacy and safety of intranasal administration of dexmedetomidine alone and in combination with ketamine as a sedative option for children who require dental sedation. We hypothesized that sedation using the intranasal combination of dexmedetomidine and ketamine is more efficacious in managing child behavior than dexmedetomidine alone.

## 2. Materials and Methods

This study is a randomized, controlled, and triple-blind clinical trial with two parallel groups (allocation ratio 1:1), and it was reported following the CONSORT guidelines [25]. The Research Ethics Committee of the Universidade Federal de Goiás (UFG), Brazil, approved the protocol. Informed consent was presented to and signed by parents/guardians for each child regarding the study’s objectives, procedures, risks, and benefits. This study was registered in the Clinical Trials database (ClinicalTrials.gov NCT03290625) before the trial starts and any patient enrollment. There were no changes in the methods after the trial began. Children recruitment started in November 2017, and the study was completed in August 2019.

### 2.1. Participants and Study Setting

The study comprised children aged 1–7 years old unable to cooperate with dental treatment (children had to show negative or definitely negative behavior, according to Frankl’s behavior rating scale [26] during a previous dental session of exam or restoration without sedation), healthy (American Society of Anesthesiologists—I; ASA) or mild systemic disease (ASA II), with reduced risk of airway obstruction (Mallampati I and II and tonsil hypertrophy occupying less than 50% of the oropharynx) [27], without neurological or cognitive disorders, those who were born term and not using medications that could impair cognitive functions. It included children with at least one decayed tooth without pulpal involvement, requiring dental restoration through atraumatic restorative treatment (ART). Children with facial deformities and who used systemic corticosteroids were also excluded. The session under sedation was scheduled at the end of the clinical examination; the interval between the child’s evaluation and the session under sedation was approximately one to two weeks.

Sedation and dental treatment were performed in an outpatient dental sedation clinic located in a university setting (Nucleo de Estudos em Sedação Odontológica’ NESO’). The NESO comprises a multi-professional team credentialled to perform dental sedation [28]: anesthesiologists, pediatricians, psychologists, pediatric and general practice dentists, graduate and undergraduate students performs dental treatment for patients of all ages with fear/anxiety or immature/deficient cognitive abilities. This team was approved and credentialled by the institution.

### 2.2. Randomization and Blinding

An external researcher generated the randomization sequence of the participants to obtain the data using an online calculator (https://www.sealedenvelope.com, Sealed Envelope Ltd., London UK). Randomization was performed considering blocks of 12 cases. For each case, the identification of the intervention group was printed and placed in a brown envelope, numbered, and subsequently sealed. The anesthesiologist secretly opened the envelope corresponding to the participant on the day of the dental treatment before sedatives’ preparation and delivery. Only pediatricians and anesthesiologists knew the group to which each child was allocated due to the need to take immediate action in case of serious adverse events. The child, his/her caregiver, the dental team (dentist and auxiliary), and the investigation observers were masked for the intervention group. During data analysis, blinding was maintained.

### 2.3. Interventions

Each participant included in the study was randomly allocated to one of the two groups. The dexmedetomidine-ketamine (DK) group received 2 mcg/kg intranasal dexmedetomidine (maximum 100 mcg) combined with 1 mg/kg ketamine (maximum 100 mg). The dexmedetomidine (D) group received 2.5 mcg/kg dexmedetomidine alone (maximum 100 mcg). The sedative formulations were ketamine injectable solution at a concentration of 50 mg/mL (Ketamin S, Cristalia^®^, São Paulo, Brazil) and dexmedetomidine injectable solution at a concentration of 100 mcg/mL (Dexmedetomidine Hydrochloride, Cristalia^®^).

On the day of the procedure under sedation, the anesthesiologists confirmed the child’s health status and fasting time [28]. At the beginning of the procedure, vital signs (heart rate, blood pressure, oxygen saturation) and the child’s body weight were evaluated and registered.

The sedatives were administered intranasally in a systematic sequence to maintain the masking of the intervention group: minute “zero”—dexmedetomidine (Groups DK and D); after 20 min—ketamine (group DK) or placebo (saline solution) (group D). Intranasal drugs or placebo were administered using an atomizer (MAD Nasal AML, San Diego, CA, USA) connected to the 1 mL syringe. In the DK group, ketamine was administered 20 min after dexmedetomidine to allow the peak plasma concentration of the two drugs to coincide. The peak plasma concentration of ketamine is reached approximately 20 min after intranasal administration [29]; the peak plasma concentration of dexmedetomidine is reached between 38 and 60 min after intranasal administration [30].

After the last drug administration, the child, caregiver, and a staff observer stayed in a dedicated space to wait for the sedatives effect. When the child reached a Ramsay score of 3 [31,32], s/he and respective caregiver were taken to the dental chair to begin the dental treatment. The child was continuously monitored by a trained observer from the administration of sedatives until discharge. The physiological data collection followed the American Society of Anesthesiologists guidelines for monitoring moderate sedation [33]. Heart rate and oxygen saturation measurements were continuously recorded during the treatment.

The dental procedures were performed by dentists who had expertise in dealing with DBMP, associating basic behavior management techniques with sedation, as demanded by the circumstances. The child’s teeth were treated using the atraumatic restorative treatment (ART) technique. In this technique, the decayed tissue is removed using manual cutting instruments, dispensing the use of rotating instruments, and, in most times, local anesthesia [34]. As many teeth were treated as possible, according to the child’s behavior. A mouth opener was used during procedures to prevent the child, sedated and drowsy, from closing the mouth. The use of the opener was not considered a form of stabilization.

The caregiver remained seated in the dental chair along with the child throughout the care. When necessary, the caregiver assisted the team in the protective stabilization, preventing disruptive movements of the children. At the end of the session, the dentist who performed the procedures classified the sedation level as minimal, moderate, or deep [32]. The entire dental care session of the child was recorded in digital media for further evaluation of their behavior. The child and respective caregiver were taken to the postanesthetic recovery area and remained there under continuous monitoring by an observer until meeting all discharge criteria according to the American Academy of Pediatrics (AAP) and American Academy of Pediatric Dentistry (AAPD) [28].

### 2.4. Outcomes

The primary outcome for this study was children’s behavior during dental sedation, which was evaluated according to the Ohio State University Behavioral Rating Scale (OSUBRS) [33]. The secondary outcomes were parental and dentist satisfaction with the dental treatment under sedation, the time needed to meet discharge criteria (recovery time), and the occurrence of adverse events.

#### 2.4.1. Primary Outcome—Children’s Behavior

The child’s behavior was assessed via digital video records from the whole appointment using the OSUBRS [35]. This scale was applied continuously and classifies child behavior into: (1) quiet behavior, no movement; (2) crying, no struggling; (3) struggling movement without crying; and (4) struggling movement with crying. The range was applied continuously throughout the digital record period using the Observer XT software (Noldus, Wageningen, The Netherlands). The percentages of time in which the child had the OSUBRS scores 1, 2, 3, and 4 were obtained.

The OSUBRS evaluations were performed by three previously trained and calibrated research assistants. Training consisted of two stages: (1) theoretical, in which each category of OSUBRS was explained and discussed, and (2) practical, in which videos of children under sedation were watched, and the behavior discussed among the evaluators and an experienced researcher. For calibration, five other videos were evaluated. The intraclass coefficients (ICC) for the interexaminer agreement were OSUBRS 1 ICC = 0.99, OSUBRS 2 ICC = 0.97, OSUBRS 3 ICC = 0.93, and OSUBRS 4 ICC = 0.99. For intraexaminer agreement, 10% of the evaluated videos were reevaluated by the same examiner. The ICC for the intraexaminer agreement were OSUBRS 1 ICC = 0.802, OSUBRS 2 ICC = 0.998, OSUBRS 3 ICC = 0.586, and OSUBRS 4 ICC = 0.999.

#### 2.4.2. Secondary Outcomes—Parents’ and Dentists’ Satisfaction; Adverse Events and Recovery Time

Parents’ satisfaction with dental sedation treatment was assessed using a 100-mm visual analog scale (VAS) [36], applied immediately after the dental treatment under sedation. Participants were instructed to record their responses, considering the anchors “completely dissatisfied” on the left and “completely satisfied” on the right. The score ranged from 0 to 100; higher scores indicated a higher level of parental satisfaction. During the registration of satisfaction with the dental treatment under sedation, the parents were left alone to avoid interference from the researcher’s presence. Dentists’ satisfaction was assessed following the same procedures [37].

Adverse events (AEs) during the transoperative and postanesthetic recovery were recorded using the TROOPS tool to be classified as minor, intermediate, or sentinel [38]. In the 24 h post-sedation period (the day after discharge), AEs were evaluated based on the caregivers’ report [39,40]. In the regression models, the presence of AEs was analyzed considering the entire session, with no distinction between the transoperative and post-discharge periods. The child’s recovery time (period from the end of dental procedure to medical discharge) was compared between groups. This outcome was not foreseen in the clinical trial protocol, but the observation of its variability led to further analysis.

### 2.5. Sample Size

The sample size was calculated based on the behavior of children during sedation for the dental procedure. As there were no studies in the dental context evaluating this variable and the sedatives in question, we sought a study related to premedication for general anesthesia [41] in which dexmedetomidine (2 mcg/kg), ketamine (2 mg/kg), an association of both sedatives (1 mcg/kg + 1 mg/kg) were compared. Analyzing the variable “ease of intravenous access,” the following success rates (good or excellent behavior) were observed: 40% in the ketamine group and 75% in the dexmedetomidine/ketamine association group. Thus, it was preliminary calculated that 36 children were necessary per group to obtain a test power of 80% and a two-tailed alpha of 0.05. The number of participants was increased by 20%, i.e., 44 children per group, to compensate for possible losses.

### 2.6. Statistical Analysis

The data were analyzed descriptively and in multiple logistic regression models, using the statistical package STATA 15.0 (StataCorp LLC, College Station, Texas, USA). Anderson-Darling and Levene tests were conducted for the primary endpoint to assess normality and variance homogeneity assumptions, respectively. An intention-to-treat approach was used for the primary analysis. Aborted cases received an OSUBRS score of 4 for children’s behavior and a score of 0 for the parent’s and dentist’s satisfaction. Missing data were handled by conditional multiple imputations (mean of 10 imputations, conditional to the trial group and child’s age).

As there was an imbalance in the age of participants in each group, probably due to the play of chance in the unstratified randomization, we compared the groups for the primary endpoint through linear regression analysis adjusted by the child’s age. The standard error was derived using a bootstrap procedure with 1000 replications to deal with possible normality assumptions deviation. The same approaches were used for the variables: parents’ and dentists’ satisfaction and recovery time. We compared the groups with logistic regression for adverse events, also adjusted by the child’s age. Sensitivity analysis with the unadjusted regression and excluding the children who had the dental procedure aborted (per-protocol analysis) were also conducted.

A one-tailed *p*-value was considered to deal with the superiority hypothesis for the primary outcome and variables related to satisfaction. A two-tailed analysis was considered to report the occurrence of adverse effects and recovery time. In all analyses, the level of significance was set at 5%.

## 3. Results

A total of 88 children (50 boys) aged between 18 and 87 months participated in this clinical trial (Figure 1). The characteristics of the participants in the baseline, dental procedures, and sedation are presented in Table 1. In group D, the children’s median age was higher than in DK. Due to this difference between the groups, primary and secondary outcomes analyses were adjusted for age.

After administering the first sedative, the time to achieve adequate sedation, defined by a Ramsay Sedation Score of 3, ranged from 32 and 59 min (median 45.0 [25th percentile–75th percentile: 39.2–50.0]). The median duration of the dental session was 25.0 [13.5–41.0]). The post-anesthetic recovery time ranged from immediately after the procedure (0 min) to 121 min (2 h) (median 45.5 [31.0–72.2]).

The planned dental treatment for the appointment under sedation was completed in 90.9% of patients in both groups (DK, *n* = 40; D, *n* = 40). In 8 (9.0%) cases (4 in group DK; 4 in group D), the restorative procedure was aborted due to the child’s uncooperative behavior. These children were included in statistical analyses (intention-to-treat analysis).

### 3.1. Child Behavior

The percentage of quiet behavior based on video analysis ranged from 0 to 100% (mean 58.1 [standard deviation 36.8]) when considering all participants. There was no significant difference in the mean of the percentage of quiet behavior between groups in the linear regression model, regardless of the children’s age (*p* = 0.225) (Table 2).

### 3.2. Parents and Operator Satisfaction during the Procedure

The mean of the parents’ satisfaction was 80.8 (standard deviation 26.1), while the mean of the dentists’ satisfaction was 65.1 (38.7). Parents and dentists were satisfied with dental treatment under sedation, without significant difference between the groups, regardless of the children’s age (parents’ satisfaction *p* = 0.203; dentists’ satisfaction *p* = 0.339) (Table 2).

### 3.3. Adverse Events (AEs)

Among the 88 children sedated, a total of 16 participants (18.2%) had an AE during the dental treatment, postanesthetic recovery (RPA), and late postoperative period. All AEs were minor, and supplemental oxygen was needed in only one case. The adverse events observed at each moment were as follows: one case of desaturation (oxygen saturation of 88%) and two cases of vomiting during dental treatment (2.3%); three cases of vomiting in RPA (3.4%); two cases of nausea, eight cases of vomiting and four cases of combined nausea and vomiting (total 14; 15.9%) in the late postoperative period. Three children with aborted procedures had AE: two cases of vomiting, one during dental treatment and the other postoperatively, and one case of nausea combined with vomiting in the postoperative period. The treatment groups were similar in the occurrence of overall AE (*p* = 0.104), regardless of children’s age (Table 2).

### 3.4. Recovery Time

Children in the DK group waited longer until medical discharge than those in group D. The mean recovery time in the DK group was 1.3 times greater than the median in group D (Table 2) (*p* < 0.05).

## 4. Discussion

In a comprehensive analysis, this study demonstrates that dexmedetomidine alone is similar to the combination of dexmedetomidine and ketamine for procedural sedation for pediatric dental procedures in the outpatient setting. Sedation with dexmedetomidine—alone or combined with ketamine—allowed the children’s behavior management during dental treatment similar to previously published studies [23,24]. Satisfactory results were observed during the treatment of caries lesions in children sedated with dexmedetomidine/fentanyl or dexmedetomidine/ketamine (all participants were calm or consolable) [24] or with dexmedetomidine only (61.6% of the children achieved successful anxiolysis) [23].

In our study, the mean duration of quiet behavior during dental treatment was similar when comparing the DK and D groups and higher than 50%. This value means that by distributing the total duration of quiet behavior evenly among children in each group, in all cases, cooperative behavior would be observed in more than half of the session duration. In the present trial, we observed less success than other investigations using dexmedetomidine for non-dental sedation. When intranasal dexmedetomidine was used at doses of 2 to 2.5 mcg/kg for transthoracic echocardiography and ophthalmic examination, success rates ranged from 85% [42] to 93.3% [43]. Differences might be due to the particularities of the procedures performed. In some medical studies, children were submitted to diagnostic tests, such as imaging, with less intense stimuli; in our study, restorative procedures with the potential to cause discomfort to children were performed. The duration of procedures also limits the comparison with medical studies since dental consultation tends to be more extensive [44,45]. In this study, the sessions lasted an average of 25.5 min, and in medical studies, the duration ranged from 7 to 13 min for ophthalmologic examination or hearing response test of the brainstem [44,45].

Parents’ and dentists’ satisfactions were secondary outcomes assessed through visual analog scales at the end of the procedure. In the context of patients’ satisfaction, the VAS is less vulnerable to bias than the Likert scale [46]. All but two participants were satisfied with the dental treatment under sedation, regardless of the treatment group. These secondary outcomes complement the assessment of the success of sedation commonly performed based on the observation of child behavior. Parents play an essential role in deciding treatment under sedation; for the child to be sedated, there must be consent from his legal guardian [47]. Thus, their satisfaction with sedation is an important benchmark when assessing sedation outcomes [37].

Similarly, the assessment of dentist satisfaction was provided in the VAS following prior studies on the satisfaction of health procedures [47,48] evaluates multiple factors related to child behavior and ease/difficulty in performing procedures under sedation. The results on the satisfaction of parents and dentists in this study, combine with the results on quiet behavior and show that sedation with dexmedetomidine, alone or combined with ketamine, is efficacious in managing child behavior in dental treatment.

Adverse events were observed in less than 14% of children; all of them were minor. This result is similar to that observed in previous studies in which children had minor AE, such as vomiting and nausea [44,49]. No major adverse events were reported.

The prolonged recovery time of children in the DK group compared to group D is noteworthy. In the first group, the recovery time was up to 1.3 times longer. This difference is significant and impacts the dental practice, throughput, scheduling, efficiency, and finances, especially in the office-based setting [42,50]. The single drug use provided a better length of recovery which is desirable for the patient and the service organization, even though the efficacy of the two sedative regimes in managing children’s behavior is similar.

Our study has limitations. Because there is a paucity of literature comparing dexmedetomidine to the dexmedetomidine-ketamine combination for pediatric sedation, it is difficult to compare our outcomes. Our study was unique because it did not evaluate success by completing the procedure [44] but instead accounted for the child’s acquiescence and behavior using observational scales. This limitation emphasizes the need to standardize the assessment of sedation success to allow greater comparability between investigations.

The success of intranasal sedation routes to provide suitable conditions for atraumatic restorative treatment encourages a pathway for future exploration. This trial supports the use of dexmedetomidine as a single agent to be administered by the intranasal route to achieve adequate sedation conditions to perform minimally invasive dental procedures in the office setting. Eliminating a second adjuvant medication (ketamine) will improve the recovery profile without altering the sedation outcomes. More extensive trials are needed to evaluate this regimen over a broader range of patient ages, psychological and medical conditions, and dental procedures.

## 5. Conclusions

The intranasal sedation with dexmedetomidine, associated or not with ketamine, is efficacious and safe for the management of child behavior in dental treatment. The DK combination prolongs post-anesthetic recovery.

## Figures and Tables

**Figure 1 jcm-10-02840-f001:**
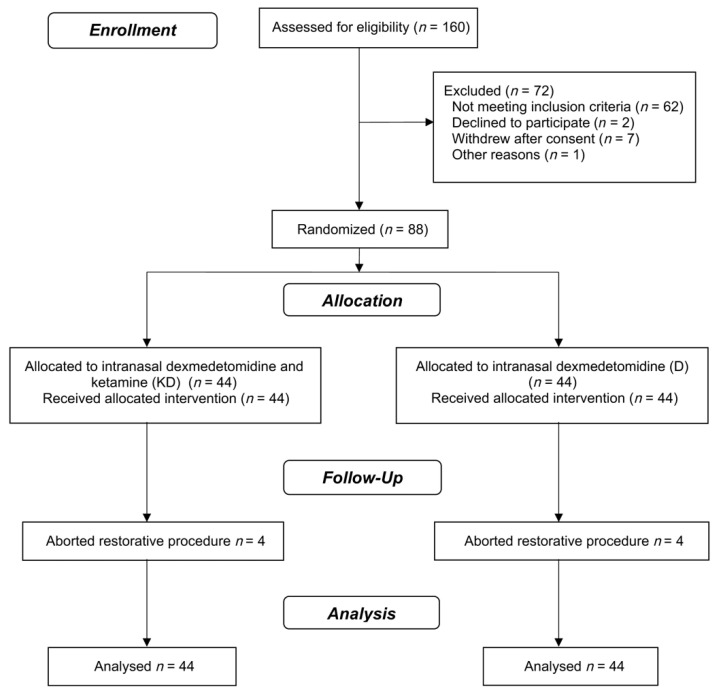
CONSORT flowchart of clinical trial progress stages.

**Table 1 jcm-10-02840-t001:** Characteristics of the participants and clinical data according to the intervention group.

Variables	Treatment Group	
	Dexmedetomidine-Ketamine	Dexmedetomidine
Participants		
Sex *n* = 88; *n* (%)		
Male	24 (48.0)	26 (52.0)
Female	20 (52.6)	18 (47.4)
Age (months) *n* = 88; median (25th percentile; 75th percentile)	36.5 (27.2–51.2)	48.0 (31.2–55.7)
Weight (kg) *n* = 88; median (25th percentile; 75th percentile)	14.5 (13.0–16.4)	16.3 (13.6–18.8)
ASA* *n* = 88; *n* (%)		
I	42 (50.0%)	42 (50.0%)
II	2 (50.0%)	2 (50.0%)
Heart rate *n* = 88; median (25th percentile; 75th percentile)	113.0 (99.0–124.0)	111.5 (99.2–119.7)
Oxygen saturation *n* = 88; median (25th percentile; 75th percentile)	98.0 (97.0–99.0)	98.0 (97.0–99.0)
Oral condition and treatment		
Caries experience *n* = 88; mean (standard deviation)	7.5 (3.5)	8.4 (3.4)
Physical restraint during the dental procedure, *n* = 80; *n* (%)		
No	17 (42.5%)	16 (40.0%)
Yes	23 (57.5%)	24 (60.0%)
Number of teeth restored *n* = 80; median (25th percentile; 75th percentile)	2 (1.2–4.7)	2 (1.0–3.0)
Sedation		
Time for onset of sedative action, median (25th percentile–75th percentile)	44.0 (38.0–47.7)	45.0 (39.0–50.0)
Duration of session under sedation, median (25th percentile–75th percentile)	25.0 (14.7–41.0)	26.0 (16.2–45.7)
Postanesthetic recovery time, median (25th percentile–75th percentile)	62.5 (41.0–77.5)	36.5 (30.0–45.5)
Adverse events		
No	32 (80.0%)	34 (85.0%)
Yes	8 (20.0%)	6 (15.0%)
Level of sedation		
Minimal sedation	13 (29.5%)	20 (45.5%)
Moderate sedation	27 (61.4%)	23 (52.3%)
Deep sedation	4 (9.1%)	1 (2.2%)

*ASA = American Society of Anesthesiologists.

**Table 2 jcm-10-02840-t002:** Comparison of outcomes observed during sedation between intervention groups (*n* = 88).

	DK	D	*p*-Value *	*p*-Value **	*p*-Value ***
	Mean (Standard Deviation)				
Primary outcome					
Quiet behavior (%)	58.4 (38.1)	55.2 (39.1)	0.225	0.35	0.33
Secondary outcomes					
Parents’ satisfaction (0–100)	78.0 (32.2)	72.7 (35.1)	0.203	0.212	0.136
Dentists’ satisfaction (0–100)	62.7 (41.0)	62.8 (40.1)	0.339	0.502	0.503
Adverse events *n* (%)			0.104	0.273	N/A
No	34 (47.2)	38 (52.8)			
Yes	10 (62.5)	6 (37.5)			
Recovery time, minutes	61.0 (25.6)	44.4 (23.6)	0.012	0.003	0.001

DK = Dexmedetomidine-ketamine; D = dexmedetomidine. * *p*-value obtained in the intention-to-treat (ITT) analysis adjusted by the child’s age (primary analysis); ** *p*-value obtained in the ITT analysis with no adjust (sensitivity analysis); *p* *** *p*-value obtained in per-protocol analysis adjusted by the child’s age (sensitivity analysis); N/A not applicable. Linear and logistic regression (significance level *p* < 0.05).

## Data Availability

The data presented in this study are available on request from the corresponding author. The data are not publicly available due to a lack of previous approval by the Research Ethics Committee.

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
