# Peer review of "Intranasal Dexmedetomidine Compared to a Combination of Intranasal Dexmedetomidine with Ketamine for Sedation of Children Requiring Dental Treatment: A Randomized Clinical Trial"

_jcm, 2021, doi:10.3390/jcm10132840_

Round 1

Reviewer 1 Report

Thank you for giving me the opportunity to review this manuscript on intranasal sedation in pediatrics, an area requiring further knowledge for better care. This project comparing IN dex to IN dex+ ket for sedation in children  was well thought and seems well realised. 

-Title: I think the title should be more modest. The superiority of Dex over Dex-Ket is not quite demonstrated in this study (the need for restrain was about 60% in both group as was the clinician satisfaction and the main objective, the quiet behavior, was reach in a mean of about 55%). I propose: Intranasal dexmedetomidine compared to a combination of Intranasal dexmedetomidine with ketamine for sedation of children requiring dental treatment: A randomized clinical 

-Line 41: It is written''Approximately four in 10 children have a dental fear/anxiety level that impedes their ability to tolerate routine, unsedated dental treatment''  However, the meta analyses cited wrote: '' Overall pooled DA prevalence was 23.9% (95% CI 20.4, 27.3). '' (DA= Dental Anxiety).  In addition, add age range of those children. Moreover, having anxiety or distress related to procedure is NOT a sine qua non of need of sedation. Non-pharmacological interventions can be effective for some of those distressed children such as distraction, deep breathing and parental presence. Please modify this sentence. See: https://www.cps.ca/en/documents/position/managing-pain-and-distress 

-Line 48-51: '' have historically been the mainstay of pediatric dental sedation associated or not with antihistaminics, meperidine, nitrous oxide, or other pharmacological agents. These sedation regimens are limited by routes of administration, long half- lives, higher risk of respiratory depression and apnea, or discontinuation of production'' -- The limitations cited do not apply to nitrous oxide. Please give information on nitrous oxide separately.

-Line 55-57: ''Ketamine offers the advantage of analgesia [15], amnesia [16], and hemodynamic stability [17] with the associated risks of apnea, respiratory compromise, laryngospasm, hallucinations, excitatory behavior, and vomiting [18,19].''--- Please adjust this sentence to the frequency of adverse events as laryngospasm is rare (less than 0.5%) and vomiting is higher (5-15%). Green 2011 https://www.annemergmed.com/article/S0196-0644(10)01827-5/fulltext should be added in the references or replace the reference 17 (Green 2004).

-Line 64-65: Maybe add https://pubmed.ncbi.nlm.nih.gov/31862730/ as a syst review on IN dex is published.

-Line 94-95: this exclusion should go with the others (line 89-90)

-Line 125: the period is missing at the end of the sentence. 

-Line 209:  there is a typo in ''to obtain a test power of 8%'': 80% ?

-Line 216: Is there a typo? '' If the procedure has been aborted due to the child's behavior, a score of 0 was defined for OSUBRS...''  As this should have been a score of 4 no?

-Line 285: This cannot be the first statement: ''In a comprehensive analysis, this study demonstrates that dexmedetomidine alone is superior to the combination of dexmedetomidine and ketamine for procedural sedation 
for pediatric dental procedures in the outpatient setting. '':

The primary outcome of this study was''...children's behavior during dental sedation, which was evaluated according to the Ohio State University Behavioral Rating Scale (OSUBRS) ...''  NOT the length of stay. In this study, the efficacy was similar, as stated in line 289. This first paragraph needs to be adjusted prior to publication. 

-Line 302: I think there is an extra ''26'' here: ''doses of 2 to 2.5 mcg/kg for transthoracic echocardiography and ophthalmic examination, success rates ranged from 85% [39] 26 to 93.3% [40].''

-Line 304-306: This publication discusses success related to painful vs non painful procedures as well: https://pediatrics.aappublications.org/content/145/1/e20191623.long

-Line 311-318: Most caregivers are satisfied with care in studies overall. I would reduce the emphasis on this result. 

-Line 323-326: This statement is too strong for a clinicians satisfaction of about 60%, and a quiet behavior of about 55%. Please adjust

-Line 329. A period is missing 

-Line 330. Maybe discuss that the children were younger in the DK group (could younger children, who take more naps, could explain part of the longer recovery period?)

-Line 348 : I would add:'' and to find the a dose or a medication to have a higher clinician's satisfaction and higher % of quiet behavior''

-Line 350: This is a strong conclusion. Please adjust. Eg: IN dex with or without ket can improve care .... 

Thanks for your contribution in pediatric sedation knowledge. 

Author Response

Reviewer: 1

Comments:

Thank you for giving me the opportunity to review this manuscript on intranasal sedation in pediatrics, an area requiring further knowledge for better care. This project comparing IN dex to IN dex+ ket for sedation in children was well thought and seems well realised.

Authors' comments: Overall, we appreciate your acceptance to revise this manuscript and your observations to improve it.

-Title: I think the title should be more modest. The superiority of Dex over Dex-Ket is not quite demonstrated in this study (the need for restrain was about 60% in both group as was the clinician satisfaction and the main objective, the quiet behavior, was reach in a mean of about 55%). I propose: Intranasal dexmedetomidine compared to a combination of Intranasal dexmedetomidine with ketamine for sedation of children requiring dental treatment: A randomized clinical

Authors' comments: We agree with the reviewer and modify the title as recommended.

1) Line 41: It is written''Approximately four in 10 children have a dental fear/anxiety level that impedes their ability to tolerate routine, unsedated dental treatment" However, the meta-analyses cited wrote:" Overall pooled DA prevalence was 23.9% (95% CI 20.4, 27.3). "(DA= Dental Anxiety). In addition, add age range of those children. Moreover, having anxiety or distress related to procedure is NOT a sine qua non of need of sedation. Non-pharmacological interventions can be effective for some of those distressed children such as distraction, deep breathing and parental presence. Please modify this sentence. See:

Authors' comments: According to the systematic review and meta-analysis cited, the pooled prevalence of dental anxiety in preschoolers was 36.5% (95%CI 23.8 to 49.2) (Grisolia et al., 2021). Overall pooled DA prevalence was 23.9% (95% CI 20.4, 27.3 refers to the grouped value for preschoolers, school children, and adolescents. In this study, children of preschool age participated; therefore, it is correct to state that approximately four in 10 children have a dental fear/anxiety.

We agree that dental anxiety is not the determining factor for the indication of sedation and that anxious children can benefit from basic behavior management techniques. However, in some cases, dental anxiety is combined with dental behavioral management problems (Klingberg, Broberg, 2007)  and, in the failure of basic techniques, sedation is indicated (AAPD, 2020). Therefore, the sentence has been revised:

"Approximately four in 10 children have dental fear/anxiety [1]. In some cases, the anxiety level is combined with dental behavior management problems and impedes the child's ability to tolerate routine, unsedated dental treatment [AAPD, 2020]. Dental anxiety and DBMP lead to an increased duration of dental procedures and elevated costs to the patient and the dental provider."

2) Line 48-51:" have historically been the mainstay of pediatric dental sedation associated or not with antihistaminics, meperidine, nitrous oxide, or other pharmacological agents. These sedation regimens are limited by routes of administration, long half- lives, higher risk of respiratory depression and apnea, or discontinuation of production"

-- The limitations cited do not apply to nitrous oxide. Please give information on nitrous oxide separately

Authors' comments: We reevaluated the paragraph and chose to remove the information about nitrous oxide. The focus of this study is sedative drugs. Therefore, information about nitrous oxide does not apply.

3) Line 55-57: "Ketamine offers the advantage of analgesia [15], amnesia [16], and hemodynamic stability [17] with the associated risks of apnea, respiratory compromise, laryngospasm, hallucinations, excitatory behavior, and vomiting [18,19]."

--- Please adjust this sentence to the frequency of adverse events as

laryngospasm is rare (less than 0.5%) and vomiting is higher (5-15%). Green 2011 https://www.annemergmed.com/article /S0196-0644(10)01827-5/fulltext should be added in the references or replace the reference 17 (Green 2004)

Authors' comments: The indicated manuscript has been read, and we have modified the sentence to state that "ketamine-associated laryngospasm is relatively uncommon (0.3% incidence in children in the large meta-analysis) (Green et al., 2011).

Ketamine offers the advantage of analgesia [15], amnesia [16], and hemodynamic stability [17] with the associated risks of apnea, respiratory compromise, hallucinations, transient laryngospasm (rare), excitatory behavior, and vomiting (more common, 5 to 15%) [Green 2011,18,19].

4) Line 64-65: Maybe add https://pubmed.ncbi.nlm.nih.gov/31862730/ as a syst review on IN dex is published.

Authors' comments: The reference has been added.

5) Line 94-95: this exclusion should go with the others (line 89-90)

Authors' comments: The sentence has been revised:

The study comprised children aged 1-7 years old unable to cooperate with dental treatment (children had to show negative or definitely negative behavior, according to Frankl's behavior rating scale,[26] during a previous dental session of exam or restoration without sedation) […]

6) Line 125: the period is missing at the end of the sentence.

Authors' comments: The sentence has been revised.

7) Line 209: there is a typo in "to obtain a test power of 8%": 80%?

Authors' comments: The sentence has been revised:

Thus, it was preliminary calculated that 36 children were necessary per group to obtain a test power of 80% and a two-tailed alpha of 0.05.

8) Line 216: Is there a typo? "If the procedure has been aborted due to the child's behavior, a score of 0 was defined for OSUBRS..." As this should have been a score of 4 no?

Authors' comments: The sentence has been revised:

" Aborted cases received an OSUBRS score of 4 for children's behavior and a score of 0 for the parent's and dentist's satisfaction."

9) Line 285: This cannot be the first statement: "In a comprehensive analysis, this study demonstrates that dexmedetomidine alone is superior to the combination of dexmedetomidine and ketamine for procedural sedation for pediatric dental procedures in the outpatient setting. ": the primary outcome of this study was "... children's behavior during dental sedation, which was evaluated according to the Ohio State University Behavioral Rating Scale (OSUBRS) ..." NOT the length of stay. In this study, the efficacy was similar, as stated in line 289. This first paragraph needs to be adjusted prior to publication.

Authors' comments We agree that the efficacy of the sedative regimens investigated is similar when assessing the primary outcome. Therefore, the paragraph was modified to highlight the primary outcome – children's behavior during dental sedation. In addition, information about recovery time has been inserted in a later paragraph.

"In a comprehensive analysis, this study demonstrates that dexmedetomidine alone is similar to the combination of dexmedetomidine and ketamine for procedural sedation for pediatric dental procedures in the outpatient setting."

10) Line 302: I think there is an extra ''26'' here: ''doses of 2 to 2.5 mcg/kg for transthoracic echocardiography and ophthalmic examination, success rates ranged from 85% [39] 26 to 93.3% [40].''

Authors' comments: The sentence has been revised.

Reviewer 2 Report

The premise of this article is sound - does a multi-drug approach work better than a single drug.  

The methods specify that a child who demonstrated negative behavior during the prior visit would then be classified as DBMP.  However, it is not clear how much time elapsed from first visit to study visit.  If six months or more elapsed between visits then cognitive maturity would play a role.  

The difference in median ages between the DK and D groups may explain the improved outcomes with a single agent.  Hence it may be that the only advantage of a single agent is reduced recovery time, since number of aborted procedures, physician and parental scores were very similar in both groups.

The evaluation of success uses an observational scale that has been used in dentistry studies by Alan Milnes in 2000 but has been supplanted by the validated Pediatric Sedation State Scale - Pediatric 2017.  The use of this scale would be preferable since it gives information not only the effectiveness of the sedation but also on how the patient tolerates the procedure.

In the conclusion, it can be stated that Dexmedetomidine has faster recovery and fewer adverse events but the results do not support stating that it is superior.

Author Response

Reviewer: 2

Comments:

1) Thank you for the opportunity to review this manuscript. The study was registered, research ethics committee approval was obtained, randomization was appropriate and power analysis was good. In the methodology I was surprised that children were administered nasal preparations are incompatible when administered together or because the authors thought that the onset of action was different between the two preparations?

Authors' comments: Overall, we appreciate your acceptance to revise this manuscript and your observations to improve it. The administration of dexmedetomidine and ketamine occurred at different times in the DK group to allow the peak moments of plasma concentration of the drugs to be close, potentiating the sedative effect. The peak plasma concentration of ketamine occurs in an average of 20 minutes after intranasal administration (Weber et al., 2004; Malinovsky et al., 1996). In turn, the peak plasma concentration of dexmedetomidine is reached within 60 minutes of administration by the same route (Iirola et al., 2011). Therefore, we added this information in the methodology:

"In the DK group, ketamine was administered 20 minutes after dexmedetomidine to allow the peak plasma concentration of the two drugs to coincide. The peak plasma concentration of ketamine is reached approximately 20 minutes after intranasal administration [29]. The peak plasma concentration of dexmedetomidine is reached between 38 and 60 minutes after intranasal administration [30]."

Weber F, Wulf H, Gruber M, Biallas R. S-ketamine and s-norketamine plasma concentrations after nasal and i.v. administration in anesthetized children. Paediatr Anaesth 2004;14(12):983-988.

Malinovsky JM, Servin F, Cozian A, Lepage JY, Pinaud M. Ketamine and norketamine plasma concentrations after i.v., nasal and rectal administration in children. Br J Anaesth 1996;77(2):203-207.

Iirola T, Vilo S, Manner T, Aantaa R, Lahtinen M, Scheinin M, Olkkola KT. Bioavailability of dexmedetomidine after intranasal administration. Eur J Clin Pharmacol 2011;67(8):825-831.

2) In the results section, because the authors have studied a group of children aged between 1 and 7 years, the age and weight should be expressed as median (IQR [range]). Also, time to onset of sedation should be expressed as median (IQR [range]). Overall, I thought this was a well conducted study and once the queries I have raised are addressed satisfactorily I would recommend publication.

Authors' comments: In table 1, age and weight are expressed in median (25th percentile; 75th percentile). Mean values of time to onset of sedation, session duration, and recovery were replaced by median (25th percentile; 75th percentile)

"After administering the first sedative, the time to achieve adequate sedation, defined by a Ramsay Sedation Score of 3, ranged from 32 and 59 minutes (median 45.0 [25th percentile - 75th percentile: 39.2-50.0]). The median duration of the dental session was 25.0 [13.5-41.0]). The post-anesthetic recovery time ranged from immediately after the procedure (0 minutes) to 121 minutes (2 hours) (median 45.5 [31.0-72.2])."

Reviewer 3 Report

Thank you for the opportunity to review this manuscript. The study was registered, research ethics committee approval was obtained, randomisation was appropriate and power analysis was good. In the methodology I was surprised that children were administered nasal preparations on two separate occasions; is this because the dexmedetomidine and ketamine preparations are incompatible when administered together or because the authors thought that the onset of action was different between the two preparations? In the results section, because the authors have studied a group of children aged between 1 and 7 years, the age and weight should be expressed as median (IQR [range]). Also, time to onset of sedation should be expressed as median (IQR [range]). Overall I thought this was a well conducted study and once the queries I have raised are addressed satisfactorily I would recommend publication.

Author Response

Reviewer: 3

Comments

1) The premise of this article is sound – does a multi-drug approach work better than a single drug. The methods specify that a child who demonstrated negative behavior during the prior visit would then be classified as DBMP. However, it is not clear how much time elapsed from first visit to study visit. If six months or more elapsed between visits, then cognitive maturity would play a role.

Authors' comments: Overall, we appreciate your acceptance to revise this manuscript and your observations to improve it. The interval between the initial consultation (clinical examination and behavior assessment) and the session under sedation was approximately one week or two weeks. Consultation under sedation was scheduled at the end of the clinical examination.  This information was inserted in the methodology.

"The session under sedation was scheduled at the end of the clinical examination; the interval between the child's evaluation and the session under sedation was approximately one to two weeks."

2) The difference in median ages between the DK and D groups may explain the improved outcomes with a single agent. Hence it may be that the only advantage of a single agent is reduced recovery time, since number of aborted procedures, physician and parental scores were very similar in both groups.

Authors' comments: Given the difference in age between the groups, we performed an analysis with adjustment for age. For behavior outcomes, physician and parental satisfaction had similar efficacy for both groups. However, a higher result for group D was observed at the recovery. This result is important and may help in the choice of sedative; however, it is not the study's primary outcome.

3) The evaluation of success uses an observational scale that has been usen in dentistry studies by Alan Milnes in 2000 but has been supplanted by the validated Pediatric Sedation State Scale – Pediatric 2017. The use of this scale would be preferable since it gives information not only the effectiveness of the sedation but also on how the patient tolerates the procedure.

Authors' comments: We appreciate your comment and suggestion of using the PSSS. In this study, we used the OSUBRS that allows the continuous evaluation of the child's behavior (quiet, crying, movement, combative). Such continuous assessment is important as children's behavior changes in the same session, depending on the sedation level, procedures, etc. Furthermore, the scale has already been used in previous publications and is valid for evaluation in the dental context:

  • Hitt JM, Corcoran T, Michienzi K, Creighton P, Heard C. An evaluation of intranasal sufentanil and dexmedetomidine for pediatric dental sedation. Pharmaceutics. 2014 Mar 21;6(1):175-84.
  • Moura LS, Costa LS, Costa LR. How do observational scales correlate the ratings of children's behavior during pediatric procedural sedation? Biomed Res Int. 2016;2016:5248271.
  • Sado-Filho J, Viana KA, Corrêa-Faria P, Costa LR, Costa PS. Randomized clinical trial on the efficacy of intranasal or oral ketamine-midazolam combinations compared to oral midazolam for outpatient pediatric sedation. PLoS One. 2019;14(3):e0213074.

4) In the conclusion, it can be stated that Dexmedetomidine has faster recovery and fewer adverse events but the results do not support starting that it is superior.

Authors' comments: We agree that it is not possible to state that dexmedetomidine is superior to the combination with ketamine. It was concluded that "intranasal sedation with dexmedetomidine alone is equally efficacious and satisfactory for pediatric sedation with fewer adverse events and faster recovery than the DK combination".